# YB-1 Knockdown Inhibits the Proliferation of Mesothelioma Cells through Multiple Mechanisms

**DOI:** 10.3390/cancers12082285

**Published:** 2020-08-14

**Authors:** Thomas G. Johnson, Karin Schelch, Kaitao Lai, Kamila A. Marzec, Marina Kennerson, Michael Grusch, Glen Reid, Andrew Burgess

**Affiliations:** 1The Asbestos Diseases Research Institute (ADRI), Concord Hospital, Concord, Sydney 2139, Australia; tjoh9110@uni.sydney.edu.au; 2The ANZAC Research Institute, Concord Repatriation General Hospital, Sydney 2139, Australia; kaitao.lai@sydney.edu.au (K.L.); kamila.marzec@sydney.edu.au (K.A.M.); marina.kennerson@sydney.edu.au (M.K.); 3Faculty of Medicine and Health, The University of Sydney Concord Clinical School, Sydney 2139, Australia; 4Sydney Catalyst Translational Research Centre, Sydney 2050, Australia; 5Institute of Cancer Research, Department of Medicine I, Medical University of Vienna, 1090 Vienna, Austria; karin.schelch@meduniwien.ac.at (K.S.); michael.grusch@meduniwien.ac.at (M.G.); 6Molecular Medicine Laboratory, Concord Repatriation General Hospital, Sydney 2139, Australia; 7Department of Pathology, The University of Otago, Dunedin 9054, New Zealand; glen.reid@otago.ac.nz; 8The Maurice Wilkins Centre, University of Otago, Dunedin 9054, New Zealand

**Keywords:** Y-box protein-1, RNA-seq, cell cycle, apoptosis, G2/M checkpoint, mitosis, p21, p53, STAT3, MDM2

## Abstract

Y-box binding protein-1 (YB-1) is a multifunctional oncoprotein that has been shown to regulate proliferation, invasion and metastasis in a variety of cancer types. We previously demonstrated that YB-1 is overexpressed in mesothelioma cells and its knockdown significantly reduces tumour cell proliferation, migration, and invasion. However, the mechanisms driving these effects are unclear. Here, we utilised an unbiased RNA-seq approach to characterise the changes to gene expression caused by loss of YB-1 knockdown in three mesothelioma cell lines (MSTO-211H, VMC23 and REN cells). Bioinformatic analysis showed that YB-1 knockdown regulated 150 common genes that were enriched for regulators of mitosis, integrins and extracellular matrix organisation. However, each cell line also displayed unique gene expression signatures, that were differentially enriched for cell death or cell cycle control. Interestingly, deregulation of STAT3 and p53-pathways were a key differential between each cell line. Using flow cytometry, apoptosis assays and single-cell time-lapse imaging, we confirmed that MSTO-211H, VMC23 and REN cells underwent either increased cell death, G1 arrest or aberrant mitotic division, respectively. In conclusion, this data indicates that YB-1 knockdown affects a core set of genes in mesothelioma cells. Loss of YB-1 causes a cascade of events that leads to reduced mesothelioma proliferation, dependent on the underlying functionality of the STAT3/p53-pathways and the genetic landscape of the cell.

## 1. Introduction

Malignant pleural mesothelioma (MPM) is an asbestos-related cancer arising from the pleural linings of the lung. MPM patients experience poor prognosis, with a five-year survival rate of only 6% [1]. Patients usually succumb to the direct extension of MPM tumours, driven by aggressive proliferation and invasion [2]. Consequently, proliferation markers such as Ki-67 are significant prognostic indicators for MPM patients, with higher scores correlating with poorer prognosis [3]. This has made blocking the proliferation of MPM cells a target of substantial interest in the development of new treatment options.

The current standard of care for MPM is a chemotherapy regime of cisplatin and pemetrexed [4], which provides an overall survival rate of only 12.1 months [5]. There are currently no effective targeted therapeutics besides bevacizumab [6] and similarly, anti-PD-1 checkpoint immunotherapy has shown limited promise in phase II trials [7]. Specifically, single-agent pembrolizumab did not improve survival compared to chemotherapy (gemcitabine or vinorelbine) in the PROMISE-Meso trial [8]. Combining anti-PD-1 inhibitors with standard of care chemotherapy did show some potential in current phase II trials however, up to 50% of patients fail to reach an objective response [8] and acquired resistance to immunotherapy drugs remains a significant hurdle to overcome [7]. Therefore, finding targeted therapeutics that block the drivers of proliferation in MPM cells could provide improved outcomes for MPM patients.

MPM is a unique cancer type, characterised by the loss of the *CDKN2A* (p16) tumour suppressor gene, which is deleted in 70–95% of MPM tumours [9,10]. Similarly, phosphatase and tensin homologue (PTEN) [9] is reportedly lost or reduced in up to 62% of MPM cases [11], while p53 is mutated in approximately 15% of cases [9]. Interestingly, there are very few known oncogenic drivers of MPM, limiting the development of targeting therapies. However, we previously demonstrated that Y-box binding protein-1 (YB-1) is commonly overexpressed in a panel of MPM cell lines compared to non-malignant mesothelial cells [12]. Furthermore, siRNA-mediated knockdown of YB-1 inhibited the proliferation, migration and invasion in 3 out of 4 MPM cell lines [12].

YB-1, encoded by the *YBX1* gene, is a multifunctional oncogene that belongs to the cold-shock domain protein superfamily. YB-1 can bind DNA, RNA and protein, leading to the regulation of a large number of cellular events, including transcription, translation, mRNA splicing, mRNA packaging, mRNA stabilisation and DNA repair [13,14]. Mutations of YB-1 are rare in MPM and other cancers (<1%), however, overexpression is strongly associated with poor prognosis [12,15]. It has been linked to multiple hallmarks of cancer including increased cellular proliferation, cell survival and invasion [16]. The mechanisms of YB-1-driven growth are multifaceted, with YB-1 regulating core cell cycle gene expression including E2F family members [17], cyclin D1 [18,19], CDC25A [20], and other proliferation genes like EGFR [21]. Additionally, YB-1 and the proto-oncogene Myc exist in a feed-forward loop in multiple myeloma, stimulating cell propagation [22]. Furthermore, YB-1 directly interacts with the critical tumour suppressor p53 [23,24], inhibiting p53-mediated apoptosis independently of p21 and MDM2 [25]. In glioma cells, YB-1 facilitates temozolomide resistance through the upregulation of MDM2 and subsequent degradation of p53 [26]. In addition to its ability to prevent p53-mediated apoptosis, YB-1 has other cell survival functions. YB-1 stimulates the pro-survival mTOR/STAT3 signalling pathway and its knockdown results in reduced STAT3 phosphorylation and MCL-1 expression [27]. It further supports this signalling by protecting STAT3 protein from proteasomal degradation but is not involved in transcriptional activation of the *STAT3* gene [28]. Additionally, YB-1 is thought to be part of the DNA repair machinery [29]. Its function as a scaffolding protein in base-excision repair [30] and its role in DNA repair of cisplatin-induced DNA damage [31] and mis-paired bases [32] further implicate it in cell survival. Taken together, it is clear that there are multiple reported mechanisms by which YB-1 could drive the proliferation of MPM. 

To determine how YB-1 controls proliferation in MPM, we utilised next-generation RNA-sequencing (RNA-seq) to characterise the gene expression changes induced by loss of YB-1 in three MPM cell lines. Knockdown of YB-1 induced both common and unique expression responses across all three cell lines, indicating that YB-1 regulates a core set of genes, which then impact MPM proliferation by different mechanisms. Notably, the p53-pathway appeared as a key central node of difference, with increased enrichment of p53-dependent gene changes correlating with cell death and/or G1 arrest while loss of p53-dependent pathways correlated with a lack of cell cycle arrest, culminating in a defective mitosis. In summary, YB-1 likely drives cell proliferation in MPM by regulating a core set of cell cycle genes, which when combined with the unique genetic background status of each cell leads to either cell cycle arrest, aberrant mitotic division and/or cell death.

## 2. Results

### 2.1. YB-1 Knockdown Alters Gene Transcription in MPM Cells 

We previously demonstrated that YB-1 knockdown can inhibit the proliferation of MPM cells in a panel of 4 cell lines [12]. To better determine the importance of YB-1 in MPM proliferation, we expanded this to 5 additional MPM cell lines. All cells were transfected with 5 nM of control (siCont) or YB-1 siRNA (siYB-1 #1) for 96 h. Of these, five out of six MPM cell lines showed a significant reduction in proliferation, with 3 cell lines reduced below 60% of siCont (Appendix A, green). We previously demonstrated that MSTO-211H, REN and VMC23 overexpress between 2.5 to 4.5-fold more YB-1 mRNA compared to non-malignant mesothelial cells and that knockdown of YB-1 inhibits VMC23 cell proliferation [12]. Based on this we selected these three cell lines for further analysis. We next validated the knockdown efficiency in each of the three cell lines by transfecting two different YB-1-specific siRNAs (siYB-1 #1 and siYB-1 #2) and compared this to siCont. Analysis of total protein confirmed both YB-1 siRNAs significantly depleted between 50–80% of YB-1 protein (Appendix A). Consistent with the above data, the proliferation of all cell lines to YB-1 knockdown was also significantly reduced by 40–60% in response to siYB-1 #1, and to a lesser extent (10–50%) with siYB-1 #2, across all 3 lines (Appendix A). Consequently, siYB-1 #1 was chosen for subsequent downstream RNA-seq analysis due to its more consistent inhibition of proliferation. To assess how YB-1 was regulating proliferation, we next performed RNA-seq analysis on each cell line that had been depleted of YB-1 or not (siCont). Initial analysis of the RNA-seq datasets revealed that over 33,000 transcripts were identified in each cell line (Appendix A). To assess the global gene expression changes between replicate samples, common dispersion of the biological coefficient of variation was measured and found to be low in all three cell lines (>0.2), implying that a significant number of genes were differentially expressed in these experiments (Appendix A). Principal component analysis confirmed that variance was primarily due to the different treatment of cells (PC1 ≥ 93%), while differences between replicates were low (PC2 ≤ 5%; Appendix A). Visual inspection of this data reflected the high reproducibility across three replicates in each cell line (Figure 1A). Two-way hierarchical clustering indicated that gene changes in MSTO-211H and VMC23 were more similar than in REN (Figure 1A), suggesting that MSTO-211H and VMC23 respond to YB-1 siRNA differently to REN. Analysis of the top 30 most significantly altered genes (up and down) showed that *YBX1* along with two of its pseudogenes *YBX1P1* and *YBX1P2* were consistently and significantly downregulated to similar levels in all three cell lines, providing a strong internal control for the YB-1 siRNA transfection (Figure 1B and Appendix A). The remaining genes in the top 30 included several notable genes linked with cell cycle and growth including *PLAC8* [33], *CCNB1*, *H2AFX, CDKN3* and *MYBL2* [34] (Figure 1B), suggesting that YB-1 promotes proliferation by regulating cell cycle-related pathways in MPM cells. 

### 2.2. Analysis of RNA-seq Data Identifies Unique Enrichment of Canonical Pathways Following YB-1 Knockdown

The above data suggested that knockdown of YB-1 inhibited MPM proliferation by deregulating gene expression of key cell cycle and growth-related pathways. To better understand YB-1-dependent effects, more detailed bioinformatic analysis of each individual cell line was conducted using Ingenuity Pathway Analysis (IPA). Using stringent cut-offs, we identified that more genes were significantly downregulated than upregulated in response to YB-1 siRNA (MSTO-211H: 955 down vs. 528 up, VMC23: 586 down vs. 395 up and REN: 903 down vs. 772 up; Figure 2A, Appendix A). As implicated by hierarchical clustering analysis (Figure 1A), each cell line displayed unique enrichments of various canonical pathways. MSTO-211H cells were significantly enriched and upregulated for G2/M checkpoint signalling, while the transcriptional activating STAT3 pathway, was significantly downregulated (Figure 2B). In VMC23, the STAT3 related Oncostatin M signalling pathway was strongly inhibited, as were GBM, ILK and to a lesser extent mitotic PLK signalling (Figure 2B). REN cells saw a strong downregulation of Rho GTPase, GP6, HOTAIR and Wnt/β-Catenin signalling and an upregulation of the STAT3 pathway (Figure 2B). All cell lines were enriched for Axonal guidance signalling, but whether the pathways were inhibited or activated could not be determined (Appendix A). Taken together, these data suggest that YB-1 knockdown induces both common and unique gene expression changes in each cell line, including several pathways that are known to regulate cell migration, proliferation and cell survival.

### 2.3. Comparative Analysis of RNA-seq Data Identifies Differential Response to YB-1 Inhibition in MPM Cells 

The above hierarchical clustering (Figure 1A) grouped VMC23 and MSTO-211H cells closer together compared to REN, which was further supported by trends observed through individual cell line analysis (Figure 2 and Appendix A). To better analyse this, we next performed comparative analysis to identify differentially enriched biological, canonical and upstream regulatory pathways. Importantly, all three cell lines showed strong inhibition of cell migration and invasion pathways (Figure 3A), supporting our previous results [12]. This was most likely driven by the loss of HOTAIR and RhoGDI signalling (Figure 3B), with both pathways implicated in the regulation of cell migration and invasion [35,36,37,38]. As implicated above, the STAT3 pathway was differentially regulated across each cell line, with inhibition seen in MSTO-211H, while VMC23 and REN cells saw increasing activity trends (Figure 3B). Interestingly, apoptosis and necrosis signals were strongly upregulated and coupled with a decrease in cell viability pathways in MSTO-211H and REN cells, but to a lesser extent in VMC23 (Figure 3A). This indicates that cell death may be contributing to the reduced proliferation observed in MSTO-211H and REN cells. Notably “cell cycle progression”, “interphase” and “mitosis” were significantly downregulated in MSTO-211H and VMC23 cells, but not in REN cells (Figure 3A), while the canonical pathway “mitotic roles of polo-like kinase” was downregulated in all cell lines (Figure 3B). Similarly, increased G2/M checkpoint signalling was seen in REN and MSTO-211H cells (Figure 3B), suggesting these cell lines were undergoing either G1 or G2/M cell cycle arrests.

To better understand these changes, analysis of potential upstream regulatory transcription, kinase and phosphatase pathways was performed. The most notable difference observed was p53-dependent transcription, which was differentially regulated across all three cell lines (Figure 3C). Specifically, significant enrichment of active upstream p53-dependent pathways was seen in MSTO-211H, a weak activation seen in VMC23, while robust downregulation of p53-dependent pathways was seen in REN cells (Figure 3C).

The upregulation of p53 transcriptional targets is commonly associated with the activation of cell cycle checkpoints and apoptosis. In support, enrichment of CDKN1A-dependent kinase signalling, which is associated with inhibition of G1 and G2 cyclin-dependent kinase (CDK) activity [34], followed a similar pattern of p53 signalling (Figure 3D). Interestingly, enrichment of downregulated E2F and upregulated CDK2NA dependent transcription, both key regulators of the G1 checkpoint [34], was observed in MSTO-211H and to a lesser extent in VMC23 and REN cells (Figure 3C). This likely explains the inhibition of FOXM1-dependent transcription, an E2F target that drives G2/M transcription, in both MSTO-211H and VMC23 cells (Figure 3C).

Notably, FOXM1 transcription was significantly decreased in REN cells despite no change to E2F (Figure 3C). In summary, the RNA-seq data indicates that each cell line undergoes a combination of G1 and G2/M cell cycle checkpoint arrests and/or cell death. Specifically, MSTO-211H cells are predicted to arrest in G1 or G2/M and undergo cell death, which correlates with a strong p53-dependent pathway response. VMC23 have a mild p53-dependent response and likely arrest in G1, while REN cells fail to instigate a strong p53-dependent response, and delay in G2/M and/or undergo cell death. 

### 2.4. Differential p53 Signalling after YB-1 Knockdown in MPM Cells 

The above upstream pathway analysis implicated p53-dependent pathways as a potential determinant regulating the response to YB-1 knockdown. Therefore, we analysed the genes responsible for the p53-dependent signature in IPA (Appendix A), which we then combined with well-known upstream regulators and downstream p53-target genes. From this we generated a short list of differentially regulated genes consisting of *MDM2, SIRT1, CDK2NA*, *CDKN1A, GADD45, IGFBP3*, *CCND1, CCNE1, CCNB1*, *MYC, E2F1* and *FOXM1* (Figure 4A–C). Interestingly, analysis of the raw expression and log fold-changes for *TP53* were significantly increased in only MSTO-211H cells following YB-1 knockdown, while both VMC23 and REN cells saw a significant decrease in *TP53* (Figure 4B,C). Of note, the total levels of *TP53* transcript were ~2–3 fold higher in control VMC23 cells compared to MSTO-211H and REN control cells, respectively (Figure 4B,C). This corresponded with a significant upregulation of the downstream targets *MDM2* and *CDKN1A* (p21), which increased in both MSTO-211H and VMC23 cells, although the increase of *CDKN1A* was not significant in VMC23 (Figure 4B,C). In contrast, REN cells expressed far lower basal levels of *CDKN1A* compared to MSTO-211H (~12 fold less) and VMC23 (~9 fold less), and these levels decreased further upon YB-1 knockdown, albeit not significantly. *CDKN2A*, which encodes the CDK4 inhibitor p16 and the MDM2 inhibitor p14^arf^, is frequently deleted in MPM [9,10]. Notably, low levels of *CDKN2A* mRNA expression were detected in MSTO-211H and REN cells, while no expression was detected in VCM23 cells (Figure 4B,C), potentially explaining the reduction in *TP53* mRNA seen in VMC23 due to loss of negative regulation of *MDM2* (Figure 4A–C). Conversely, the transcription factor *MYC* (c-Myc), which negatively regulates p14^arf^, was significantly downregulated in MSTO-211H cells (Figure 4B,C), potentially explaining the increase in *MDM2* in these cells. A significant decrease in the G1 cyclin *CCND1* (cyclin D1) and its transcription factor *E2F1*, was observed in all three cell lines (Figure 4A). Similarly, the G2/M cyclin *CCNB1* (cyclin B1) and its transcription factor *FOXM1*, were significantly reduced in all three lines, while *GADD45A*, which is downstream of p53 and regulates the G2/M checkpoint, was significantly reduced in MSTO-211H and REN cells (Figure 4B,C). Interestingly, expression of *CCNE1* (cyclin E1), which drives S-phase, was significantly increased in both MSTO-211H and VMC23 cells but remained unchanged in REN cells. *IGFBP3*, which promotes p53-dependent apoptosis [39,40], was significantly up in MSTO-211H cells, but down in both VMC23 and REN cells. Conversely, *SIRT1*, which deacetylates p53 and negatively regulates the activity of downstream pro-apoptotic BAX, was significantly increased in both VMC23 and REN cells, but not in MSTO-211H cells (Figure 4B,C). 

One potential explanation for the variable changes in p53 signalling could be due to potential mutations in TP53, which occur in ~15% of MPM tumour samples [9]. MSTO-211H have previously been annotated as wild-type for p53 [41], however, the status of REN and VMC23 cells has not yet been published. We therefore utilised our RNA-seq data and performed alignment and SNV variant calling for the *TP53* gene in each cell line and cross-referenced this with the IARC TP53 (version R20) database [42]. Interestingly, we identified two variants in exon 4 in MSTO-211H (c.215C>G, and c.348T>C), which resulted in the common P72R missense polymorphism and a silent mutation at S116S (Figure 4D, Appendix A). VMC23, also contained the missense P72R polymorphism, along with three other silent mutations (A63, G187, R196). In contrast, REN cells did not carry the P72R mutation, but did carry two polymorphisms in exon 11 (c.314G>A, c.409C>A) of unknown significance, along with 22 intronic SNVs mostly in intron 1, the majority of which are validated polymorphisms with no known clinical significance (Appendix A). Consequently, based on these data, it is likely that all three cell lines express wild-type p53, with MSTO-211H and VMC23 both containing the P72R polymorphism.

To further examine the p53 pathway and validate the RNA-seq data, we analysed total protein levels of YB-1, p53, p21 and MDM2 in each cell line with or without YB-1 knockdown. Both YB-1-specific siRNAs significantly depleted YB-1 protein by ~80% across all three cell lines (Figure 4E and Appendix A). In correlation with the RNA-seq data (Figure 4B), p21 levels significantly increased in MSTO-211H cells following YB-1#1 knockdown (Figure 4E), which corresponded with a reduction in MDM2 levels and an increase in a lower molecular weight band, while total p53 levels did not significantly change. In contrast, VMC23 cells showed no significant changes in p21 levels, which corresponded with stable MDM2 and low but stable expression of p53. Finally, REN cells had much lower levels of p21 protein, which reduced further upon YB-1 depletion (Figure 4E), similar to the RNA-seq data. Surprisingly, this was despite higher levels of basal p53 and lower basal MDM2 levels. Notably, all three lines expressed detectable p53 protein of the expected size, supporting the SNV alignment analysis that all three lines contained wild type p53.

In summary, these data suggest that MSTO-211H and VMC23 cells likely undergo p53-dependent cell cycle arrests. However, in MSTO-211H cells, increased expression of IGFBP3 potentially primes cells for apoptosis, while loss of IGFBP3 combined with increased SIRT1 may protect VMC23 cells from death.

In contrast, REN cells, despite expressing p53 protein, contain very low levels of p21, which does not get upregulated upon loss of YB-1, suggesting that the p53 pathway may be partially dysfunctional in these cells. 

### 2.5. Identifying Potential Core Target Genes of YB-1

The above analysis indicated that the block in proliferation in each cell line was caused by a combination of cell cycle checkpoint arrest and induction of cell death. To better understand how YB-1 was inducing these effects, we next analysed our RNA-seq data to determine whether there were any downstream YB-1 targets common to all three cell lines. Of the genes upregulated following YB-1 knockdown, only 3.1% (42 in total) were found to be increased in all three cell lines (Appendix A). STRING analysis identified only two clusters consisting of two connected nodes each, suggesting that genes suppressed by YB-1 were not likely to contribute significantly to the proliferation of MPM cells (Appendix A). 

In contrast, there were 150 common downregulated genes, representing 8.3% of the total (Figure 5A). Importantly, STRING analysis identified a core network of 117 genes (Figure 5B), which were significantly enriched for the “polo-like kinase-mediated events” (FDR = 1.8 × 10^−3^) and “cell cycle, mitotic” (6.5 × 10^−3^) Reactome pathways (Figure 5B,C). Similarly, the top three Biological Processes (GO) hits were “regulation of nuclear division”, “regulation of mitotic nuclear division” and “mitotic cell cycle process” (Appendix A). We next compared our data to previous studies by Kwon et al. [43] and Li et al. [44]. There were no common genes upregulated across all 3 studies, however, there were 24 and 4 genes from the Kwon and Li studies, respectively, that overlapped with our study (Appendix A), although these genes were not enriched for any known pathways. In contrast, there was a total of 89 genes from Kwon or Li that were also downregulated in our dataset (Appendix A). STRING analysis of the commonly downregulated genes (with Kwon, Li or both) identified significant enrichment of genes involved in regulating cell adhesion and migration (e.g., *MCAM, MMP2*), DNA replication and repair (e.g., *CDC6, FACNB, RAD51AP1*) and mitosis (e.g., *AURKB, CDC20, KIF18B*), actin/myosin regulation (e.g., *CNN1, TPM2*) and smaller networks regulating Wnt and Activin signalling (e.g., *FST, INHBA*) (Appendix A). Taken together, these data suggest that YB-1 may drive, either directly or indirectly, genes that are essential for cell migration, proliferation and cell cycle (DNA replication, repair and mitosis), providing a possible explanation for the reduced proliferation observed in MPM cells.

### 2.6. Growth Inhibition after YB-1 Knockdown Occurs via Apoptosis or an Altered Cell Cycle 

The above RNA-seq data suggested that a combination of either cell cycle arrest and/or cell death were driving the reduced proliferation phenotypes observed in MSTO-211H, VMC23 and REN cells. 

To confirm this, TALI^™^ apoptosis assays, which utilise Annexin V Alexa Fluor 488 (An) and propidium iodide (PI) to determine necrotic (An-/PI+), early (An+/PI-) and late apoptotic cells (An+/PI+), were performed on each cell line 96 h after transfection with 5 nM YB-1 or control siRNA. MSTO-211H cells had a significant increase in late apoptotic cells and a decrease in viable cells (An-/PI-) after YB-1 knockdown (Figure 6A), confirming the enrichment of cell death signals seen by RNA-seq in Figure 3A. In VMC23, there was no significant increase in apoptotic or necrotic cells, nor a decrease of viable cells in response to YB-1 siRNA transfection compared to the control transfected group (Figure 6A). This again supported the reduction of *IGFBP3* and increase in *SIRT1* observed in the RNA-seq data (Figure 4B) and the upstream pathway analysis (Figure 3A) that predicted an inhibition of cell death and increased anti-apoptotic signalling in VMC23 cells. Finally, REN cells displayed a significant increase in necrotic cells and a decrease in viable cells (Figure 6A), suggesting that REN undergo cell death, but not necessarily classical apoptosis, in response to YB-1 siRNA. Taken together these results confirm the above RNA-seq data that predicted higher rates of cell death in MSTO-211H and REN compared to VMC23 cells. The RNA-seq data also implied that all 3 cell lines underwent various cell cycle arrests. To identify how YB-1 knockdown could be affecting the cell cycle, DNA content flow cytometry on cells transfected with 5 nM of YB-1 or control siRNA was conducted 96 h after transfection. Concordant with TALI^™^ and RNA-seq results, YB-1 silencing induced a significant increase in the sub-G0 (dead cell) population for MSTO-211H and REN cells, compared to control, but not for VMC23 (Figure 6B). All three cell lines displayed a significant decrease in the G2-M population (Figure 6B), while VMC23 and REN cells showed a significant increase in G0/G1 cells (Figure 6B). REN also displayed a significant decrease in S phase (Figure 6B). These data indicate that all three cell lines were likely delayed in G0/G1 at 96 h after YB-1 knockdown, supporting the RNA-seq and p53/p21 data for VMC23 and MSTO-211H cells, however, this does not explain the predicted G2/M arrest in REN cells.

### 2.7. Live-Cell Imaging Confirms that YB-1 Knockdown Induces Three Different Growth Inhibitory Phenotypes 

The above static, single end-point flow cytometry data was unable to provide a clear validation for the various cell cycle arrests predicted by RNA-seq. Therefore, to better analyse the cell cycle in greater detail, live-cell video microscopy combined with single-cell fate mapping was conducted on each cell line, as previously described [45]. 

Briefly, asynchronously growing cells transfected with control (siCont) or YB-1 siRNA, were monitored every 10 min for 72 h one day after transfection and a minimum of 50 cells per treatment were scored for cell cycle length (interphase + mitosis), mitotic fidelity and cell fate (Figure 7A). 

Under control siRNA conditions, both MSTO-211H and REN cells proliferated and died at a higher rate than VMC23. Knockdown of YB-1 reduced the number of divisions and exacerbated cell death in both MSTO-211H and REN cells, while VMC23 remained largely refractory to cell death, as predicted by the RNA-seq data. These observations were confirmed by quantifying the total number of divisions each cell performed, with the average number of divisions significantly reduced from 2 to 1 in MSTO-211H and VMC23, and from 3 down to 1 in REN cells after YB-1 knockdown (Figure 7B). This corresponded with a significant increase in interphase length in VMC23 (+492.1 min), REN (+627.3 min). MSTO-211H cells also showed an increase in interphase time (+283.1 min), although this increase was not significant, likely due to the large increase in cell death (Figure 7C). Interestingly, in REN cells, mitotic length was significantly increased with cells taking on average 127.95 min longer to divide compared to controls (Figure 7D). To better understand the mechanism of cell death and arrest, we next quantified the phenotypes of cells that arrested before (ABM) or after mitosis (AAM), or cells that died before (DBM), during (DDM) or after (DAM) a mitotic division (Figure 8A). Cells were considered to have a normal baseline phenotype if they divided twice or more (2+) over the 72-h period (Figure 8A). YB-1 knockdown significantly reduced cells undergoing a normal cell cycle by 38%, 42% and 56% in MSTO-211H, VMC23 and REN cells, respectively (Figure 8B). In MSTO-211H cells, a clear increase in cells dying before a division and arresting after a mitotic division was observed (Figure 8B), suggesting that cells that divide arrest in G1 in a p53-p21 dependent checkpoint. Similarly, the proportion of VMC23 cells that arrested after a division doubled to 44%, however, instead of dying during interphase, the number of cells that never underwent a division increased from 4% to 22% (Figure 8B). In contrast, the number of REN cells that died during a prolonged mitosis (DDM) or arrested after mitosis increased by 24% and 34%, respectively (Figure 8B). This corresponded with a significant increase in cells undergoing an aberrant cytokinesis characterised by multiple cleavage furrows, which either subsequently survived or underwent cell death (Appendix A). Taken together, it is clear that in all three cell lines there was a notable increase in the percentage of cells arresting after mitosis. For MSTO-211H and VMC23 cells, the interphase arrest was likely driven by p53 signalling. However, as REN cells likely do not have a functional p53-p21 pathway response, their ability to arrest in G1/G2 was reduced, allowing them to continue through the cell cycle and complete a defective mitosis, leading to either mitotic catastrophe or aberrant cytokinesis, which likely caused the daughter cells to arrest.

## 3. Discussion

Over-expression of YB-1 has long been associated with driving cancer proliferation, migration and invasion in a number of cancer types, including breast [46], colon [47], lung and MPM [13]. Conversely, knockdown/out of YB-1 has been shown to block the proliferation and migration of a variety of cancer cell types including MPM [12,48]. However, over-expression has also been shown to reverse AKT-induced oncogenesis [49], while in non-cancerous H9C2 cells (rat cardiomyocytes), loss of YB-1 promotes cellular proliferation [50]. This suggests that the role of YB-1 in regulating proliferation in cancer is likely dependent on the individual cell type and its genomic and mutational landscape. In support, our data demonstrated that while loss of YB-1 blocked the proliferation of 3 MPM cell lines, the phenotypic mechanisms for this reduced proliferation were unique to each cell line. Specifically, MSTO-211H cells were highly sensitive to apoptosis and cell death, VMC23 cells underwent a stable G1 arrest, while, REN cells underwent a prolonged mitosis resulting in mitotic catastrophe or subsequent G1 cell cycle arrest. 

Analysis of the gene expression changes found that similar to previous reports, loss of YB-1 resulted in the downregulation of genes that control DNA replication, mitosis and cell migration. These included some known targets of YB-1, such as CDC6 [51], but also several genes, such as PLK1 and Aurora B, that have not been previously associated with regulation by YB-1. Notably, YB-3 (gene *YBX3,*
Appendix A), which has been reported to substitute for YB-1 in its absence and thereby minimise the effects of YB-1 loss [52], was not upregulated in any of our YB-1-depleted cell lines. Consequently, without YB-3 expression, these cells may be unable to compensate for the loss of YB-1, potentially explaining the large number of gene expression changes that we observed. Despite a similar set of core gene expression changes occurring in each cell line, there were noticeable differences in how each cell responded. Specifically, enrichment of STAT3 and p53-dependent pathways were key differentiating factors amongst the three cell lines. Intriguingly, there was an inverse relationship between STAT3 and p53 pathway activation, with MSTO cells seeing loss of STAT3 and increased p53-pathway activity. The STAT3 pathway is commonly activated in cancer, driving cell proliferation and metastasis [53]. Notably, STAT3 can repress p53 [54], while conversely the p53 target p21 can repress STAT3 activity [55], likely explaining the inverse relationship observed. Interestingly, YB-1 has been previously shown to directly interact with p53 [23], reducing its levels and transcriptional activity. Consequently, loss of YB-1 has been shown to increase p53 activity and induce p53-dependent apoptosis [24]. Our results correlated partially with these previous observations, with YB-1 knockdown in MSTO-211H cells significantly increasing p53 mRNA levels, which subsequently resulted in increased p21 mRNA and protein. In addition, MSTO-211H cells also contained the P72R polymorphism, which has been shown to correlate with increased apoptosis, while P72 correlates with stronger cell cycle arrest [56,57,58,59]. Consequently, the presence of R72 in MSTO-211H may explain the predisposition for apoptosis in these cells. However, neither REN nor VMC23 cells followed the same pattern, with p53 mRNA levels and activity decreasing, as noted by the failure to induce p21 levels, suggesting that loss of YB-1 was not sufficient to drive p53 activity in these two cell lines. These results indicate that the phenotypes we observed cannot be solely dependent on the direct interaction between YB-1 and p53. Similarly, the P72R polymorphism, which is present in VMC23 but not REN cells, also does not explain the lack of cell death seen in VMC23 cells, or the reduced cell cycle arrest observed in REN cells. Clearly, other factors beyond p53 itself, must also impact the response to YB-1 knockdown. 

Expanding the p53 network to include other known regulators and mediators of p53 function, generated a clearer picture of why VMC23 and REN cells behaved differently to MSTO-211H cells. Specifically, both REN and VMC23 saw significant increases in *SIRT1* and reduced *IGFBP3* mRNA, which likely reduces the triggering of apoptosis in these cells. Notably, in contrast to a previous report in glioma cells [26], we saw a significant increase rather than decrease in *MDM2* mRNA following YB-1 knockdown in both MSTO-211H and VMC23 cells, while REN cells showed no significant change. Interestingly, the primer used by Tong et al. to detect MDM2 in glioma cells [26] maps to exon 8, which is commonly spliced out in various MDM2 isoforms. Intriguingly, we observed decreased MDM2 protein and a lower molecular weight form, despite increased mRNA in MSTO-211H. YB-1 has been shown to regulate alternative splicing of *MDM2* transcripts [60], with knockdown of YB-1 promoting exon skipping resulting in less stable MDM2 protein, potentially explaining the reduced MDM2 in MSTO-211H cells. Gaining a better understanding of the complex interplay between YB-1, p53 and MDM in MPM will be of future interest. 

REN cells were unique and characterised by a failure to upregulate p53, with both mRNA and protein decreasing after YB-1 knockdown, indicating that REN cells, despite likely containing wild-type p53, are unable to upregulate p21 expression. This likely explains why these cells were unable to sustain stable G1 or G2 arrests after YB-1 knockdown. This resulted in cells eventually attempting to complete the cell cycle, culminating in a prolonged, aberrant mitosis with cells either dying through mitotic catastrophe or undergoing a defective cytokinesis and arresting in the following interphase. YB-1 has previously been reported to bind to pericentrin and γ-tubulin during mitosis, where it regulates centrosome-based microtubule nucleation and organisation. Notably, over-expression or loss of YB-1 has been previously shown to disrupt the mitotic spindle and cause cytokinesis defects [61,62,63]. Interestingly, all three cell lines saw significant downregulation of several key S-phase and mitotic regulators such as CDC6, H2AX, PLK1, cyclin B2, Aurora B, TACC3, KIF20A and CDC20. This was most likely an indirect consequence due to the loss of E2F and FOXM1 driven S/G2 transcriptional activity seen across all three cell lines, particularly as E2F1 is transcriptionally upregulated by YB-1 [17]. However, it may be of future interest to determine if any of these are direct YB-1 targets as loss of any of these genes may also prevent the successful completion of DNA replication and promote aberrant mitotic division.

## 4. Materials and Methods 

### 4.1. Cell Culture

Cell lines used in this study included MSTO-211H, H226 (both bought from the ATCC, Manassas, VA, USA), VMC20, VMC23 (kindly provided by the Medical University of Vienna, Vienna, Austria), MMO5 (kindly provided by The Prince Charles Hospital, Brisbane, Australia) and REN (kindly provided by The University of Pennsylvania Medical Centre, Pennsylvania, PA, USA). Cells were maintained in Roswell Park Memorial Institute-1640-GlutaMAX^™^ (RPMI) medium supplemented with 10% heat-inactivated foetal bovine serum (ThermoFisher Scientific, Waltham, MA, USA) in a humidified atmosphere (5% CO_2_ and 37 °C). Cells were routinely monitored for mycoplasma and their identity confirmed by short tandem repeat profiling (Australian Genome Research Facility, Melbourne, Australia). 

### 4.2. Transfection of siRNA

YB-1 and validated negative control siRNA were purchased from Shanghai GenePharma (Shanghai, China) using the following antisense sequences: YB-1 #1 (UUUGCUGGUAAUUGCG UGGAGGACC), YB-1 #2 (UAUUUCUUCUUGUUGGAUGACUAAA), validated negative control (AAGCAACUUGGUAAGACUCGUGUGG). Cells were reverse transfected using 0.1% Lipofectamine RNAiMAX (ThermoFisher Scientific) as per the manufacturer’s protocol.

### 4.3. RNA Isolation, RNA Sequencing (RNA-seq), SNV Alignment and Analysis

Cells (1 × 10^5^) were reverse transfected with 5 nM of control or YB-1-specific siRNA. RNA was collected 96 h later using the TRIzol extraction kit (Thermo Fisher) as per manufacturers guidelines. DNA was removed using a TURBO DNA-free Kit^™^ (Invitrogen by Thermo Fisher Scientific, Carlsbad CA, USA) as per the manufacturer’s guidelines. RNA concentration and 260/280 ratio were determined on an Implen nanophotometre^™^ (Implen, München, Germany). Samples were only used if they showed a 260/280 ratio >1.8. RNA integrity was determined on a 2100 Bioanalyser (Agilent Technologies, Santa Clara, CA, USA) and samples were only used if they showed a RIN of >8. Three sets of RNA were collected per cell line and per treatment. Compliant samples were sent to the Australian Genome Research Facility (AGRF) for RNA sequencing with poly(A) selection. Briefly, 20 million 100 bp single end RNA-seq was conducted on a NovaSeq platform (Illumina, San Diego, CA, USA). The library was prepared using the TruSeq stranded RNA sample preparation as per the manufactures protocol (Illumina, San Diego, CA, USA). The cleaned sequence reads were aligned against the Homo sapiens genome (Build version hg38) and the STAR aligner (v2.5.3a) was used to map reads to the genomic sequence. Transcripts were assembled using the StringTie tool v1.3.3 with the read alignment (hg38) and reference annotation-based assembly option (RABT). Raw data were deposited in the NCBI Gene Expression Omnibus (GEO) data repository accession number GSE153368.

The raw data from each cell line was aligned to the human genome reference build GRCh38/hg38 using RNA-seq aligner “Spliced Transcripts Alignment to a Reference (STAR)” v2.5.3a by AGRF. Single Nucleotide Variations (SNVs) were identified using SNV caller Freebayes (v1.3.1) and annotated using Bcftools (v1.9) with database NCBI dbSNP (v146). Heatmaps, principal component analysis and biological coefficient variant plots were made using R software (The R Foundation, Vienna, Austria), using the DESeq2 package. The log2 (fold change) scale has been normalised and transformed by considering library size or other normalisation factors. The transformation method and the variance-stabilising transformation (VST) [64] for over dispersed counts have been applied in DESeq2. The VST is effective at stabilising variance, because it considers the differences in size factors, such as the datasets with large variation in sequencing depth [65]. Canonical Pathway analysis of known proliferation, cell cycle, migration and cell death-related signalling pathways were conducted using the Ingenuity Pathway Analysis software (QIAGEN). Minimum significance cut-offs of *p*-value > 0.05 and Z scores of >2 and <−2 were applied for pathways analysis. Volcano and dot plots were made using GraphPad PRISM (v8.4.2, GraphPad Software, San Diego, CA, USA). Venn diagrams were generated using Venny 2.1.0 [66] and figures compiled using Adobe Illustrator (v24.1.3, San Jose, CA, USA).

### 4.4. Protein Isolation and Western Blot

After 96 h, cells transfected with 5 nM Control or *YBX1* siRNA were collected and protein isolated by lysis in radioimmunoprecipitation assay (RIPA) buffer supplemented with protease inhibitor cocktail (ThermoFisher Scientific). Protein concentration was measured using a Pierce bicinchoninic acid assay (ThermoFisher Scientific). Proteins were separated by sodium dodecyl sulphate polyacrylamide gel electrophoresis and blotted onto PVDF or nitrocellulose membranes. Immunodetection was performed using specific antibodies (monoclonal rabbit anti-YB-1 (ab12148, Abcam; 1:1000), rabbit monoclonal anti-p53 (2527S, Cell Signaling Technology, Danvers, MA, USA; 1:1000), monoclonal rabbit anti-p21 (2947S, Cell Signaling Technology; 1:1000), and mouse anti-MDM2 (ab16895, Abcam, Cambridge, UK; 1:200) and Bio-Rad Clarity (Hercules, CA, USA) enhanced chemiluminescence (ECL). Membranes were then exposed to film or signal images captured using a Bio-Rad ChemiDoc XRS+ System. β-actin (antibody A2228, Sigma Aldrich, St. Louis, MO, USA; 1:1000) or GAPDH (G9545, Sigma Aldrich, 1:5000) served as loading controls. Densitometry was performed using Fiji [67] or Bio-Rad Image Lab software and a minimum of three biological repeats were performed.

### 4.5. SYBR Green-Based Cell Proliferation Assays

Cells (2.5 × 10^3^ per well) were reverse transfected with siRNAs at the indicated concentrations in a total volume of 120 μL per well in 96-well plates. Plates were frozen at −80 °C after 96 h. Plates were then thawed, lysis buffer (10 mM Tris/HCl pH = 8, 2.5 mM EDTA, 0.1% Triton X-100) containing SYBR green (10,000×, ThermoFisher Scientific, 1:8000) was added and incubated overnight at 4 °C. Fluorescence was read on a FLUOstar OPTIMA microplate reader (BMG Labtech, Ortenberg, Germany) at 485/535 nm. Experiments were performed three times in triplicates.

### 4.6. Apoptosis Assays

Cells (1 × 10^5^) were transfected with 5 nM of negative control or YB-1-specific siRNA in 6-well plates as described above in a total volume of 2.4 mL per well and incubated at 37 °C and 5% CO2 for 96 h. Detached cells were collected in the supernatant, attached cells were detached using trypsin, all were pelleted at 1500 rpm for 5 min, washed with PBS and pelleted again. These live cell populations were stained using a TALI^™^ Apoptosis Kit (ThermoFisher Scientific) which included Annexin V Alexa Fluor^™^ 488 (An) and propidium iodide (PI), as per manufacturers guidelines. Slides were read on a TALI^™^ Image-Based Cytometer (ThermoFisher Scientific) using 18 fields of view, as per manufacturer’s guidelines. Experiments were repeated three times.

### 4.7. Cell Flow Cytometry Cell Cycle Analysis

Cells (1 × 10^5^) were transfected with 5 nM of control or YB-1-specific siRNA in 6-well plates as described above in a total volume of 2.4 mL per well and incubated at 37 °C and 5% CO2 for 96 h. Cells were detached using trypsin, pelleted at 1,500 rpm for 5 min, washed with PBS and pelleted again. Cells were fixed in 70% ethanol and left at 4 °C for 2–5 days. Fixed cells were pelleted, ethanol aspirated and cells resuspended in a solution of 15 μg/mL propidium iodide (Sigma) and 12.5 μg/mL RNAse A (ThermoFisher) in PBS for 1 h at 37 °C. Cell flow cytometry was conducted on a BD Accuri^™^ C6 Flow Cytometer (BD Biosciences, Franklin Lakes, NJ, USA) and analysed on FlowJo using the Dean-Jett-Fox model. Experiments were repeated three times.

### 4.8. Live-Cell Imaging and Cell Fate Analysis

Cells (1 × 10^5^) were transfected with 5 nM of control or YB-1-specific siRNA in 6-well plates as described above in a total volume of 2.4 mL per well and incubated overnight at 37 °C and 5% CO_2_. The next morning, 2.5 × 10^4^ of these cells were seeded into 12-well plates in RPMI +10% FBS for live-cell imaging. Phase-contrast videomicroscopy was conducted using the 10× Plan Fluorite NA 0.25 objective for 72 h with photo intervals of 10 min using an EVOS FL Auto Live Cell Imager (ThermoFisher). Manual tracking and cell fate mapping were performed as previously described [45]. Experiments were repeated a minimum of two times, with 50 individual cells counted for each treatment in each cell line.

### 4.9. Statistical Analysis

Data are presented as mean plus or minus standard error of the mean. Unless otherwise stated statistical analysis was conducted on Prism 8.4.2 software. For analysis of two groups, a two-tailed unpaired t-test was conducted. For analysis of three or more groups, a two-way ANOVA test with a Sidak’s multiple comparison test was conducted. Unless otherwise stated all *p* values less than 0.05 were considered statistically significant.

## 5. Conclusions

In summary, we have demonstrated through RNA-seq that knockdown of YB-1 in MPM cells drives a large number of gene expression changes, many of which have not previously been reported. This dataset, which may be of significant value to the wider research community, identified unique phenotypic responses across three MPM cell lines, which was responsible for the observed reduction in cell proliferation. Loss of YB-1 drove the downregulation of a core set of common key genes involved in cell cycle and migration, which, combined with the overall functionality of the STAT3 and p53 pathways, determined if cells were able to initiate a stable cell cycle arrest, trigger apoptosis or fail to arrest and progress through an aberrant mitosis. Consequently, a complete understanding of the genetic background and functionality of the p53 and STAT3 pathways will be important for any future therapeutic interventions that target YB-1.

## Figures and Tables

**Figure 1 cancers-12-02285-f001:**
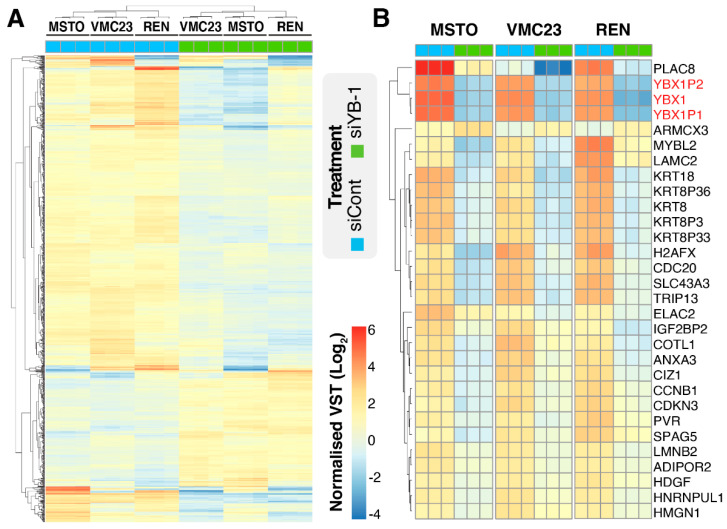
Transfection of YB-1 siRNA downregulates several core cell cycle gene transcripts. (**A**) Heatmap of all altered genes identified by RNA-seq and (**B**) of the top 30 altered genes of RNA-seq data of MSTO-211H, VMC23 or REN cells transfected with siCont or siYB-1 #1, based on significance (FDR). Experiments were repeated three times. Heatmaps with two-way hierarchical clustering were generated using R software. The heatmap colour spectrum depicts variance-stabilising transformation (VST) counts relative to the mean value of all VST counts of all samples. Highlighted in red are *YBX1* and the two pseudogenes, *YBX1P1* and *YBX1P2*.

**Figure 2 cancers-12-02285-f002:**
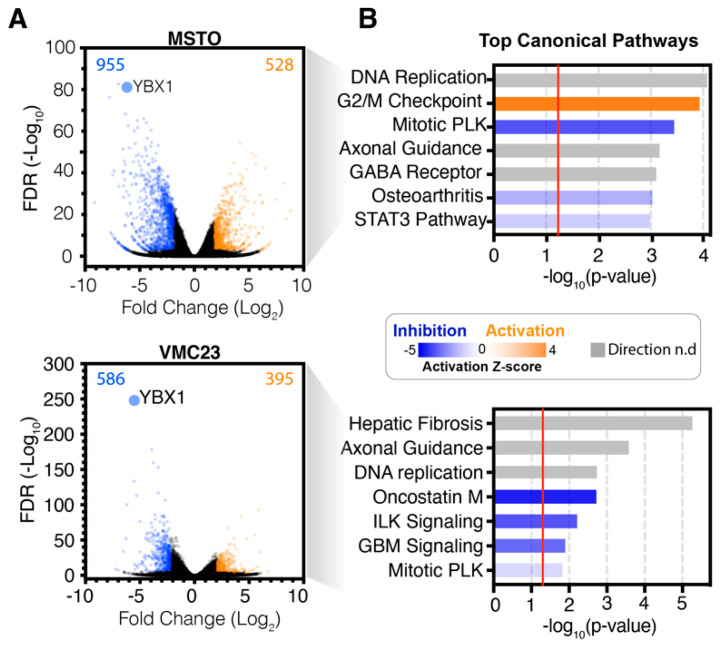
RNA-seq analysis identifies that proliferation-related pathways are altered in MPM cells after YB-1 siRNA transfection. (**A**) Volcano plots displaying significantly downregulated (blue) or upregulated (orange) genes after YB-1 knockdown. Significance cut-offs were defined as log_2_(fold change) < −2 and > 2, *q*-value (FDR) < 0.001. (**B**) IPA canonical pathway analysis of the RNA-seq data. Predictions of inhibition (blue) or activation (orange) or no change (white) states are based on the Ingenuity^®^ Knowledge Base, which compares the expected change with experimental observation to all known upstream regulators. Variable stringent *p*-value (>1.3) and z-score (>0.5) cut-offs were used to limit pathways to top 7–8 most significant hits. Grey bars indicate where no directionality could be determined (Direction n.d).

**Figure 3 cancers-12-02285-f003:**
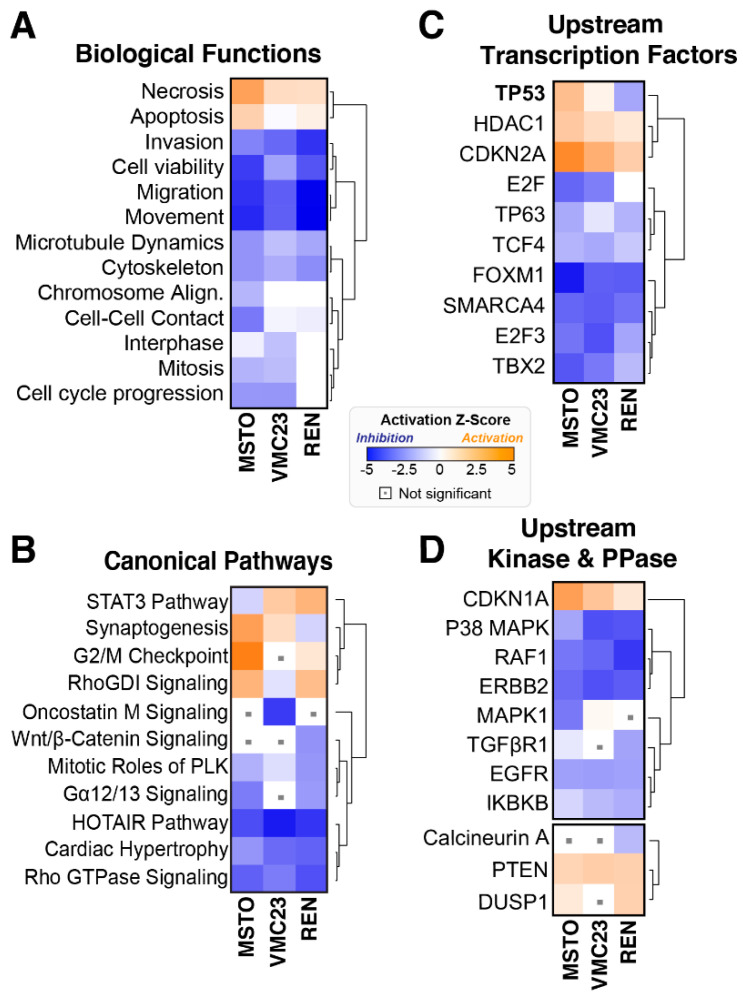
Comparative analysis of RNA-seq data suggests MPM cells respond differently to YB-1 knockdown. (**A**–**D**) Hierarchical clustering of IPA comparative analysis for indicated pathways. Variable stringent *p*-value (>1.3) and z-score (>0.5) cut-offs were used to limit pathways to below top 15 most significant hits. Orange (activation), blue (inhibition), white (no change), insignificance threshold < 0.05 (grey dot).

**Figure 4 cancers-12-02285-f004:**
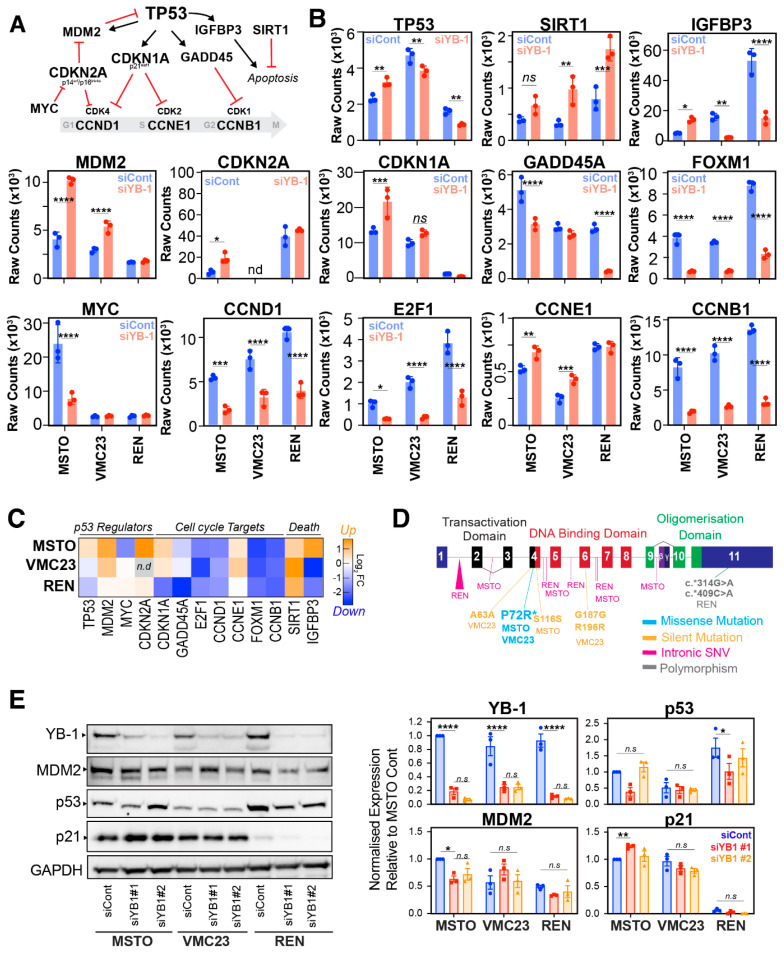
Analysis of the p53 signalling network in YB-1 depleted MPM cells. (**A**) Schematic of major regulators and downstream targets of the p53 pathway. (**B**) Raw counts from RNA-seq analysis for individual genes involved in the p53 pathway across the 3 MPM cell lines. (**C**) IPA based comparative analysis of log-fold (Log_2_FC) expression changes for p53-pathway genes. MSTO-211H = MSTO. (**D**) Summary of mutational analysis of p53 using IGV view to map RNA-Seq reads to *TP53* reference gene. (**E**) Immunoblots of whole protein harvested from MPM cells 96 h after transfection with 5 nM of either control (siCont) or YB-1 siRNA (siYB-1 #1 and #2). Western blot analysis of total protein for YB-1, MDM2, p53, p21, and GAPDH (loading control). * *p* < 0.05, ** *p* < 0.01, *** *p* < 0.001, **** *p* < 0.0001, not significant (n.s).

**Figure 5 cancers-12-02285-f005:**
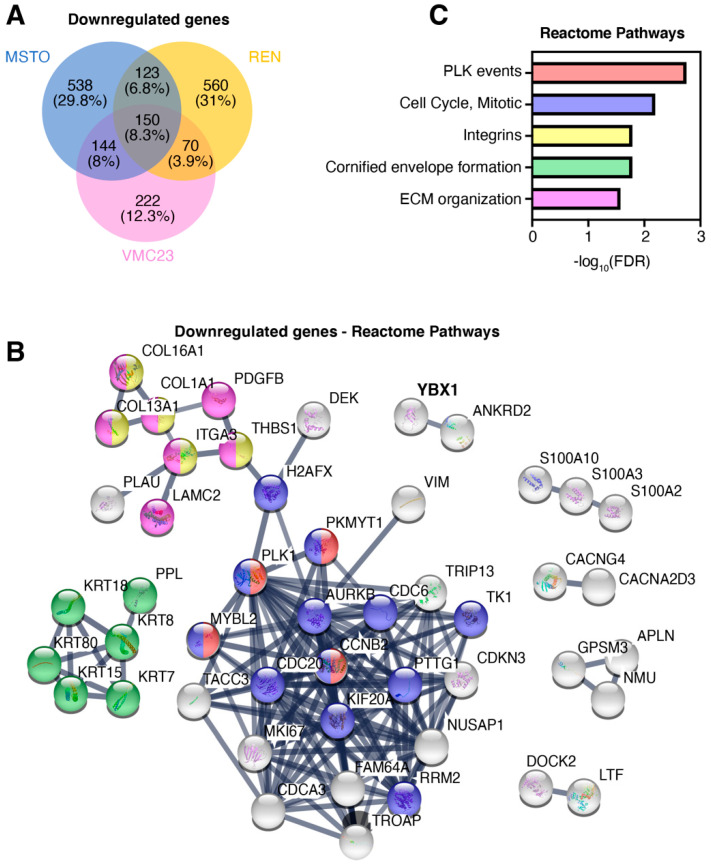
Downregulated genes common to all three cell lines are associated with the mitotic cell cycle. (**A**) Venn diagram of all significantly downregulated genes (log_2_(fold change) < −2; *q*-value (FDR) < 0.001) in each cell line, displayed as number of genes (percent of total genes across three cell lines). (**B**) STRING-DB Reactome Pathway analysis of common downregulated human genes (*n* = 150) across all three cell lines. Network edges are displayed as ‘confidence’ and disconnected nodes are hidden. Genes highlighted in colour correspond to (**C**) the top five most significant Reactome pathways. Significance is shown as −log_10_(FDR).

**Figure 6 cancers-12-02285-f006:**
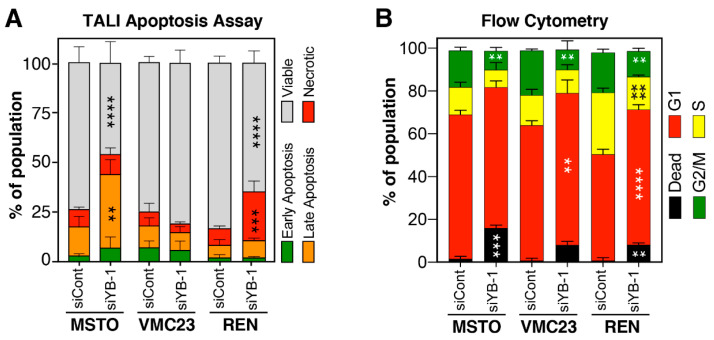
YB-1 knockdown in MPM cells induces apoptosis or cell cycle alterations. Cells were transfected with 5 nM Control (siCont) or siYB-1 #1 and analysed 96 h post transfection with (**A**) TALI apoptosis assays, which utilise Annexin V Alexa Fluor™ (An) and propidium iodide (PI) to identify viable (An-/PI-), early apoptotic cells (An+/PI-), late apoptotic cells (An+/PI+), necrotic cells (An-/PI+) or (**B**) DNA content flow cytometry assays. Dead cells (sub-G0, black), G1 (red), S (yellow) and G2/M (green). ** *p* < 0.01, *** *p* < 0.001, **** *p* < 0.0001. All data shown are from (*n* = 3) biological repeats presented as a mean +/− standard deviation.

**Figure 7 cancers-12-02285-f007:**
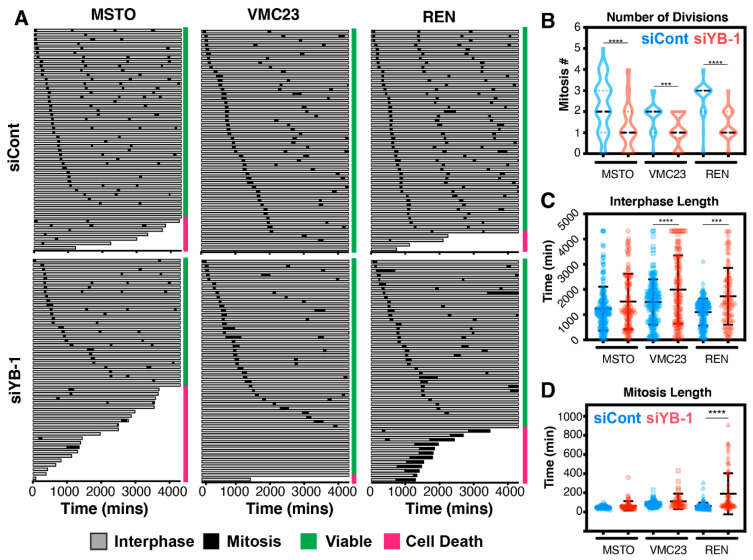
Multigenerational cell fate mapping analysis reveals three distinct phenotypic changes in MPM cells after YB-1 knockdown. (**A**) Multi-generational cell fate maps of 50 individual cells from three MPM cells transfected with 5 nM Control (siCont) or YB-1 siRNA #1 (siYB-1). Cells were transfected, incubated overnight and seeded for videos, which ran for 72 h. Time spent in interphase (grey), mitosis (black), viable cells (green bar) and cells that die (pink bar) are shown. (**B**) Violin plots of total number of divisions within in the 72-h period. Mean divisions (black dotted bar). (**C**,**D**) Dot plots of total time (mins) spent in interphase (grey bars) and mitosis (black bars) for each individual cell across each MPM cell line shown in (**A**). *** *p* < 0.001, **** *p* < 0.0001.

**Figure 8 cancers-12-02285-f008:**
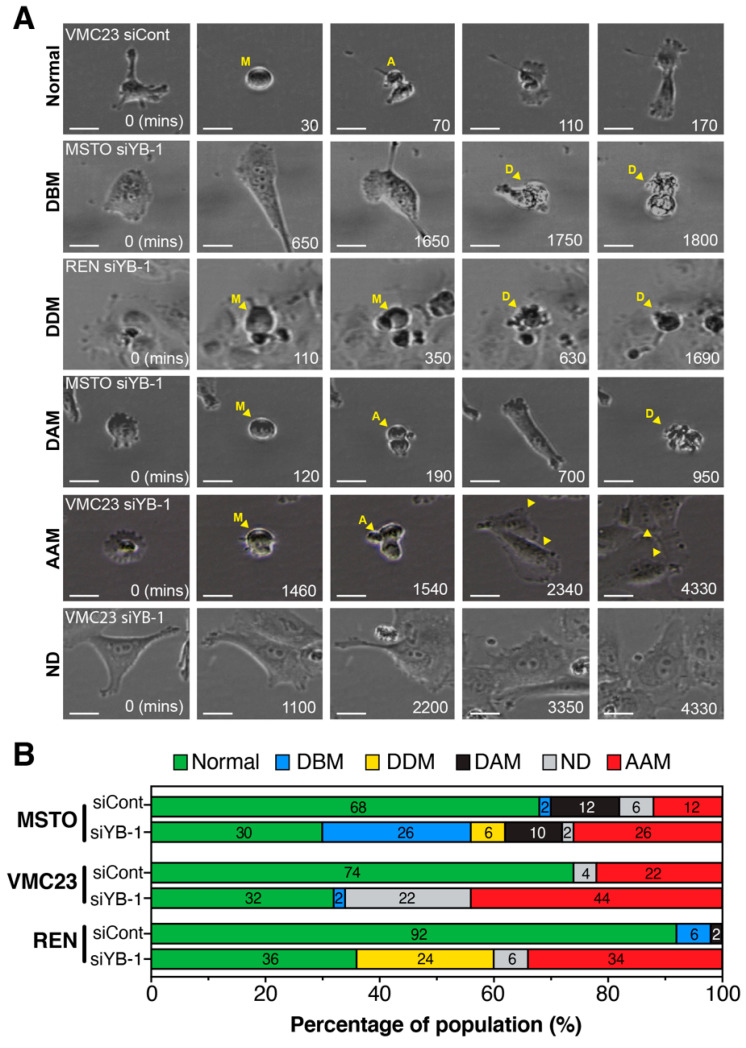
Live cell imaging of YB-1 knockdown phenotypes (**A**) Representative images and (**B**) quantification of cells transfected with either Control (siCont) or YB-1 siRNA (siYB-1 #1) monitored for 72 h (time in min) by time-lapse microscopy and scored based on observed phenotype. Normal (2+) = cells that underwent two or more mitoses and remained viable. DBM = cells that died before undergoing any mitosis. DDM = cells that died during mitosis. DAM = cells that died after a mitosis. AAM = cells that arrested after mitosis and did not divide again. No division (ND) = cells that did not divide at all during 72 h. Times (bottom right) are in minutes. M = mitosis, A = anaphase, D = cell death. Scale bars = 20 µm.

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
