# Peer review of "YB-1 Knockdown Inhibits the Proliferation of Mesothelioma Cells through Multiple Mechanisms"

_cancers, 2020, doi:10.3390/cancers12082285_

Round 1

Reviewer 1 Report

In the current manuscript Johnson et al. analyzed the changes in transcriptomic profile of three different malignant pleural mesothelioma (MPM) cell lines upon YB-1 knockdown. Previously authors demonstrated that knock-down of YB-1 inhibits proliferation, migration and invasion of MPM cells. In the current study authors aimed to identify the mechanism underlying this effect. They performed RNA-sequencing and analyzed changes in gene expression caused by YB-1 knock-down. Authors were able to identify common and specific changes in gene expression among three MPM cell lines. Also, using time-lapse video microscopy authors demonstrated that know-down of YB-1 impairs cellular division and induces cell cycle arrest.

Expression of YB-1 is often dysregulated in cancer and the question addressed in this work is of interest to the field. Targeting YB-1 may provide an efficient approach to treat patient with mesothelioma. Nevertheless, I am not convinced that the authors provided a significant mechanistic advance in the deciphering of how YB-1 controls proliferation of MPM cells.

Major points:

  • Throughout the text authors say that they analyze transcriptional changes induced by YB-1 knock-down. However, the role of YB-1 in mRNA stability is well known (see for example Yang, 2019, Evdokimova, 2001), and its demise can affect the stability of cellular transcriptome. In agreement with this, the results obtained in the current work show that more genes were down-regulated than up-regulated. Moreover, role of YB-1 as a transcription factor has been recently challenged (Dolfini, 2013). How do the authors distinguish at which level YB-1 regulates gene expression in MPM cells?
  • Do MPM cell lines express higher levels of YB-1 as compared to non-cancerous cells? In the introduction authors speculate that YB-1 can be a target of hyperactive PI3K/AKT/mTOR signaling pathway. It would be nice if authors analyze the phosphorylation status of YB-1 in MPM cells.
  • It is not completely clear why authors focus on p53 expression. In fact, the changes of TP53 mRNA levels are quite modest in MSTO cell line (~0.5 fold increase) (Figure 4B). Moreover, the level of p53 protein is decreased when MSTO cells are treated with siYB1A#1 (Figure 4E) that was used for RNA-seq. Finally, authors do not provide any experimental evidences of how YB-1 regulates the expression of p53.
  • Authors should explain better why did they chose to compare MSTO-211H, VMC23 and REN cell lines, introducing their origin and differences in genetic background.

Minor points:

  • Line 186. Authors should explain why HOTAIR and RhoGDI signaling inhibits cell migration and invasion, and provide corresponding references.

Author Response

I would like to thank the reviewer for their helpful and insightful comments, which have significantly improved the manuscript.

Point-by point response:

Q1) Throughout the text authors say that they analyze transcriptional changes induced by YB-1 knock-down. However, the role of YB-1 in mRNA stability is well known (see for example Yang, 2019, Evdokimova, 2001), and its demise can affect the stability of cellular transcriptome. In agreement with this, the results obtained in the current work show that more genes were down-regulated than up-regulated. Moreover, role of YB-1 as a transcription factor has been recently challenged (Dolfini, 2013). How do the authors distinguish at which level YB-1 regulates gene expression in MPM cells?

A1) The reviewer is correct and we have altered the text throughout the manuscript to make it clear that the gene expression changes oberseved with YB-1 knockdown could be due to a number of possibilities, including mRNA stability or transcription changes. This includes adding in the references suggested by the reviewer, improvements to the introduction to highlight the additional functions of YB-1 and removing references to transcription from results and the discussion.

Q2) “Do MPM cell lines express higher levels of YB-1 as compared to non-cancerous cells... 
and, "Authors should explain better why did they chose to compare MSTO-211H, VMC23 and REN cell lines, introducing their origin and differences in genetic background?"

A2) This is an important point, and consequently, we have now included 5 additional MPM cell lines and combined this with our previous publication (Johnson, JTO, 2018) where we demonstrated that YB-1 is commonly over-expressed compared to the normal, immortalized mesothelial cell line MeT-5A. Importantly, we provide a clearer rationale for why each cell line was chosen, based on the level of YB-1 overexpression and effect of knockdown on proliferation (lines 103-109). This new data has been included in Supplementary Figure S1A.

Q3) In the introduction authors speculate that YB-1 can be a target of hyperactive PI3K/AKT/mTOR signaling pathway. It would be nice if authors analyze the phosphorylation status of YB-1 in MPM cells.

A3) We agree that the points raised in the introduction are misleading with regards to this study. Unfortunately, knockdown of YB-1 protein prevented us from accurately analysing the AKT-dependent phosphorylation changes on YB-1. Consequently, we have removed the section on AKT in introduction to better reflect the content of the manuscript.

Q4) It is not completely clear why authors focus on p53 expression. In fact, the changes of TP53 mRNA levels are quite modest in MSTO cell line (~0.5 fold increase) (Figure 4B). Moreover, the level of p53 protein is decreased when MSTO cells are treated with siYB1A#1 (Figure 4E) that was used for RNA-seq. Finally, authors do not provide any experimental evidences of how YB-1 regulates the expression of p53.

A4) We agree with the reviewer, this section was not clearly written and confusing. We have improved the analysis of this section and included an additional supplementary table (Table S2), that covers all of the specific genes deregulated in our dataset that are downstream of the p53-nework, as identified by our IPA analysis. We have also significantly re-written the sections on Figure 2, 3 and 4 to more clearly explain that it is the p53-pathway that is differentially deregulated, not necessarily p53 itself. We believe that this now clarifies the confusion surrounding this point.

Q5) Authors should explain why HOTAIR and RhoGDI signaling inhibits cell migration and invasion and provide corresponding references.

A5) We have now added additional references [35-38] (lines 162-163) to better explain how HOTAIR and RhoGDI signalling are related to cell migration and invasion.

  1. Cai, B.; Song, X.Q.; Cai, J.P.; Zhang, S. HOTAIR: a cancer-related long non-coding RNA. Neoplasma 2014, 61, 379-391, doi:10.4149/neo_2014_075.
  2. Liu, M.; Zhang, H.; Li, Y.; Wang, R.; Li, Y.; Zhang, H.; Ren, D.; Liu, H.; Kang, C.; Chen, J. HOTAIR, a long noncoding RNA, is a marker of abnormal cell cycle regulation in lung cancer. Cancer Sci 2018, 109, 2717-2733, doi:10.1111/cas.13745.
  3. Cho, H.J.; Kim, J.T.; Baek, K.E.; Kim, B.Y.; Lee, H.G. Regulation of Rho GTPases by RhoGDIs in Human Cancers. Cells 2019, 8, doi:10.3390/cells8091037.
  4. Hodge, R.G.; Ridley, A.J. Regulating Rho GTPases and their regulators. Nat Rev Mol Cell Biol 2016, 17, 496-510, doi:10.1038/nrm.2016.67.

Reviewer 2 Report

This is a well-written paper and I have only a few minor comments:

Figure 1:  Where the heatmap colour spectrum depicts 'log fold-change', what is this change relative to?  If the answer is 'relative to control', what is the colour of the fold-change for the control - shown on the same plot - relative to?  Or are these both relative to some median point?

Figure 2B:  If the three subfigures representing the different cell lines use different criteria for choosing pathways, but your argument compares them, then you should include the non-activated pathways in each subfigure as well.  For instance, if the activated pathways for MSTO (top subfigure) do not include 'hepatic fibrosis', but the activated pathways for VMC23 do include this pathway using different criteria, then it should be included in the MSTO subfigure.  This will show that the difference in which pathways are activated isn't just a consequence of the different criteria.  Because this will probably make the figure pretty unwieldy, it could be done as a supplementary figure.  

l.125, l.489:  'principle component analysis' should read 'principal component analysis'. 

l.213:  'p53-dendent' should read 'p53-dependent'. 

l.296:  I feel that this statement: 'Importantly, our study identified a greater number of up and downregulated genes compared to either the Kwon et al or Li et al studies, highlighting the potential resource of our dataset to the research community.' is a bit of a stretch.  If these studies were not conducted using the same cell lines, the same siRNAs, and the same fold-change cutoffs for significance, then the number of up and down-regulated transcripts isn't really a measure of the comparative value of any study. 

l.397: The sentence '...potentially explaining why we saw significantly more transcriptional changes than this report.' is confusing to me.  Can you rephrase this?

Author Response

I would like to thank the reviewer for their helpful and insightful comments, which have significantly improved the manuscript.

Q1) Where the heatmap colour spectrum depicts 'log fold-change', what is this change relative to?  If the answer is 'relative to control', what is the colour of the fold-change for the control - shown on the same plot - relative to?  Or are these both relative to some median point?

A1) We have corrected this error and included additional details in the figure legend and methods section to better explain the colours within the heatmap. Specifically, within the legend we have included the text “Heatmaps with two-way hierarchical clustering were generated using R software. The heatmap colour spectrum depicts variance-stabilising transformation (VST) counts relative to the mean value of all VST counts of all samples.”

The heatmap key within the figure has also been relabelled to Normalised VST (Log2), and the method section includes the following text: “The log2 (fold change) scale has been normalised and transformed by considering library size or other normalisation factors. The transformation method, the variance-stabilising transformation (VST) [64], for over-dispersed counts have been applied in DESeq2. The VST is effective at stabilising variance, because it considers the differences in size factors, such as the datasets with large variation in sequencing depth.” (lines 481-485).

Q2) If the three subfigures representing the different cell lines use different criteria for choosing pathways, but your argument compares them, then you should include the non-activated pathways in each subfigure as well.  For instance, if the activated pathways for MSTO (top subfigure) do not include 'hepatic fibrosis', but the activated pathways for VMC23 do include this pathway using different criteria, then it should be included in the MSTO subfigure. This will show that the difference in which pathways are activated isn't just a consequence of the different criteria. Because this will probably make the figure pretty unwieldy, it could be done as a supplementary figure. 

A2) As requested we have included an additional figure (Figure S3) that combines all of the pathways analysed across each cell line. In addition, we have significantly improved the text surrounding Figure 2 and Figure 3 to more clearly explain the analysis and to distinguish between the individual cell line analysis performed in Figure 2 and the comparative analysis performed in Figure 3.

Minor text changes:

A) We have corrected 'principle component analysis' to read 'principal component analysis'

B) 'p53-dendent' has been corrected to read 'p53-dependent'

C) We have removed the statement comparing the number of genes between the Kwon and Li studies with ours.

D) We rephrased the sentence “…potentially explaining why we saw significantly more transcriptional changes than this report”, with the following text to add greater clarity.

 “thereby minimising the effects of YB-1 loss [46], was not upregulated in any of our YB-1-depleted cell lines. Consequently, the lack of YB-3 expression in the cell lines we analysed suggests that these cells are less able to compensate for YB-1 loss, potentially explaining the large number gene expression changes that we observed.”

Round 2

Reviewer 1 Report

I believe that authors significantly improved the manuscript and I suggest to accept in the present form. The datasets generated in this study will be of interest to scientific community.